Functional outcomes and quality of life after a 6-month early intervention program for oral cancer survivors: a single-arm clinical trial

Chen Yueh-Hsia 1
Liang Wei-An 1
Hsu Chung-Yin 2
Guo Siang-Lan 1
Lien Shwu-Huei 1
Tseng Hsiao-Jung 3
Chao Yuan-Hung yuanhungchao@ntu.edu.tw 2 4 5
1 Rehabilitation Center, Department of Plastic and Reconstructive Surgery, Chang Gung Memorial Hospital Linkou Branch , Taoyuan , Taiwan
2 School and Graduate Institute of Physical Therapy, College of Medicine, National Taiwan University , Taipei , Taiwan
3 Center for Big Data Analytics and Statistics, Chang Gung Memorial Hospital Linkou Branch , Taoyuan , Taiwan
4 Rehabilitation Center, National Taiwan University Hospital Chu-Tung Branch , Hsinchu County , Taiwan
5 Center of Physical Therapy, National Taiwan University Hospital , Taipei , Taiwan
Brown Chris
Electronic publication date: 2018 Feb 21
Publication date: 2018
Volume: 6
Electronic Location ID: e4419
Received 2017 Jul 18; Accepted 2018 Feb 6
Copyright: ©2018 Chen et al.
Copyright year: 2018
Copyright holder: Chen et al.
License: This is an open access article distributed under the terms of the Creative Commons Attribution License, which permits unrestricted use, distribution, reproduction and adaptation in any medium and for any purpose provided that it is properly attributed. For attribution, the original author(s), title, publication source (PeerJ) and either DOI or URL of the article must be cited.
License URL: https://creativecommons.org/licenses/by/4.0/

Keywords: Oral cancer, Return to work, Shoulder dysfunction, Quality of life

Funding: Chang Gung Medical Research Program CMPRG3E0611 This study was financially supported by the Chang Gung Medical Research Program (CMPRG3E0611), Tauyuan, Taiwan. The funders had no role in study design, data collection and analysis, decision to publish, or preparation of the manuscript.

==============================
Background

Advanced treatment of oral cancer increases survival rates; however, it also increases the risk of developing shoulder dysfunction, dysphagia, oral dysfunction, donor site morbidity and psychological issues. This single-arm preliminary pilot study aims to explore the effects of a six-month early intervention program following reconstructive surgery in oral cancer survivors.

Methods

A total of 65 participants were analyzed following reconstructive surgery. Outcome measurements were taken during the first visit, and at one, three and six months after reconstructive surgery.

Results

Scapular muscle strength and shoulder range of motion progressively improved during the 6-month follow-up. The mean Disability of the Arms, Shoulder and Hand (DASH) score showed significant improvement at 1 month (p < .001). Health related QoL showed significant differences between baseline and 6-months post-surgery scores on global health and on most of the function and symptom scales. The predicted return-to-work rate was 80% at one year after the operation. Return-to-work rate differs in different vocational types, with a higher rate of return in the skilled or semi-skilled (87.5%) and self-employed (86.7%).

Conclusions

We suggest that early integrated intervention program with a follow-up of at least six months following reconstructive surgery may help develop and identify intervention guidelines and goals in the initial six months of treatment following neck dissection in oral cancer survivors.

Introduction

Oral cavity cancer is the 11th most common cancer in the world. Its incidence rate is highly correlated with exposure to tobacco, betel nut chewing, and alcohol in developing nations, and the incidence of human papillomavirus (HPV) infection in developed countries (Krishna Rao et al., 2013; Marur & Forastiere, 2016; Sankaranarayanan et al., 2015). Advanced surgical technology in oral cancer increases the survival rate, but the primary functions of the oral cavity—respiration, speech, mastication, deglutition, and appearance—are significantly compromised. Impairments include scar contracture or radiation-induced trismus, drooling, impaired mastication and deglutition with a lip or tongue defect. In the advanced stage of cancer, donor-site-related impairments often develop, such as accessory nerve shoulder dysfunction (ANSD) (Cappiello et al., 2005; Dijkstra et al., 2001; Stuiver et al., 2008; Van Wilgen et al., 2003). The probability of shoulder dysfunction after neck dissection is as high as 70% (Carr, Bowyer & Cox, 2009; Dijkstra et al., 2001). ANSD is manifested as impaired shoulder mobility and pain. Electromyography (EMG) studies have shown significant increase in spinal accessory nerve denervation (Erisen et al., 2004) and decreased trapezius muscle activity after neck dissection (Lima, Amar & Lehn, 2011; McGarvey et al., 2013a). Though the modified radical and selective neck dissections aims to reduce the prevalence of shoulder dysfunction, a wide range of ANSD incidence rates after neck dissection have been reported (Sheikh, Shallwani & Ghaffar, 2014; Umeda et al., 2010).

Health-related quality of life (QoL) has been extensively studied in oral cancer survivors to identify the impact of the treatment-related morbidity and physical issues. This is used for newly diagnosed oral cancer as well as recurrence. Recent studies have demonstrated that poor oral and physical health-related QoL are found in oral cancer patients compared with the general population (Barrios et al., 2015). Shoulder dysfunction (decreased muscle strength, range of motion [ROM] and pain) is associated with decreased QoL (McNeely et al., 2015). Postoperative radiation therapy is associated with decreased global health, increased xerostomia and short term fatigue (Ch’ng et al., 2014). Issues of oral function, such as swallowing, speech and social eating ensue one year or more after the reconstructive surgery. Poor role function and social function are predicted in the advanced-stage oral cancer survivors as well (Schliephake & Jamil, 2002). In these, shoulder dysfunction, global health, oral function and social function would be the aim of address in this study.

Given the issues mentioned above, this study aims to explore changes in physical function and self-perceived QoL in consequence of an early intervention program after oral cancer reconstructive surgery. While most research focuses on a single measure of treatment outcome and its effect on QoL, we assumed that a range of treatment outcome measures would affect health-related quality of life, including symptoms, dysfunctions, and survival. Therefore, this research aims to evaluate the outcomes of an early-intervention program (i.e., progressive resistance exercise, soft-tissue massage, joint range of motion exercise, pain management and functional training) by physical examination, QoL questionnaire and return-to-work status after oral cancer reconstructive microsurgery through exploratory analyses. Through exploring the outcomes in the early and the advanced stage population eligible for surgery, the results may help develop and identify intervention guidelines and goals in the initial 6 months of treatment following neck dissection in oral cancer survivors.

Materials and Methods

This study was a single-arm preliminary pilot study. The participants were enrolled from January 2015 to June 2016. The inclusion criteria were survival after excision of oral cavity squamous cell carcinoma (SCC) with reconstructive microsurgery, and age between 20 and 65 years old. Patients are recruited through the plastic surgeon’s referral. Patients were excluded if they were identified as cognitively impaired; having distant metastasis or recurrence; or unable to communicate or comprehend the questionnaires. Written informed consent and verbal trial information was provided and obtained from all participants. Informed consent for publication of identifying images of study participants was also obtained. This study was conducted in the Chang Gung Memorial Hospital in Taoyuan, Taiwan, and approved by the Chang Gung Medical Foundation Institutional Review Board (Approval No: 103-5164B, 104-2300C, 104-8154C). The authors declare that all methods were performed according to the relevant guidelines and regulations established by the oversight boards and agencies.

Interventions started early after the reconstructive microsurgery. All participants underwent each component of the intervention program, which consisted of pain management, scar management, temporomandibular joint (TMJ) exercise, shoulder and neck exercise, and functional training of the donor site and recipient site to restore oral and physical function. The intervention program was divided into three phases: (1) the early phase (within 1 month after operation), (2) the middle phase (1 to 3 months post-operation), (3) and the late phase (more than three months post-operation).

Early phase

The early phase intervention initiated at an average of 8.3 days following reconstructive surgery and lasted within one month post operation. In the early phase, the main goal was to help the participants deal with the problems suffered (i.e., pain, edema, shoulder dysfunction and strength loss, soft tissue tightness and functional limitation) as the result of the surgery. The intervention consisted were as follows: (1) Transcutaneous Electrical Stimulation was administered for 15 min. per treatment session and the intensity was adjusted to patient’s tolerance to alleviate shoulder pain or soreness. (2) Gentle soft tissue mobilization and scar massage on the donor and recipient sites was performed for 15 min. per treatment session and the intensity was adjusted to patient’s tolerance to prevent edema-induced stiffness. (3) To increase TMJ range of motion, the participants were guided to exercise their TMJ during the first visit and to use tongue depressors to assist in performing the TMJ exercise if needed in the following visits. (4) Active or active-assistive exercises for neck, shoulder and donor site were performed for 10 repetitions in each session. (5) The progressive shoulder resistance exercise consisted of closed-chain and open-chain exercises that were adjusted sequentially according to pain conditions (Fig. 1). (6) In cases involving fibular osteoseptocutaneous flap harvest, big toe flexion was avoided in the initial six weeks. (7) Functional training such as transfer, ambulation, and activities of daily living (ADL) was administered before discharge from the hospital. All of these were initiated as soon as the wound condition was stable (i.e., normal capillary refill decided by the surgeon clinically.) The participants were instructed to perform self-massage and other ROM exercises hourly during waking hours and was checked using exercise diaries kept by the participants. During the period of hospitalization (average duration = 3 weeks), participants received the intervention every weekday for an average of 40 min. per treatment session. The treatment program was administered once a week after discharge from hospital.

Figure 1 Scapula control exercise.

(A–C) Anti-scapular winging exercise: bilateral hands held together and elevated to the top then placed behind the head, pulling the bilateral scapular closer. (D) Wall press exercise. (E) Wall press with scapular elevation exercise. (F) Wall press with arm clock exercise.

Middle phase

The middle phase lasted from 1–3 months post operation. During this phase, the goal was to focus on the impairment suffered (i.e., scar development, soft tissue tightness, shoulder joint dysfunction and strength loss, functional limitation) from the surgery or radiation therapy (RT). The participants attended the rehabilitation center for one-on-one therapy once per week. The interventions included: (1) Scar massage was employed 5 to 10 min prior to other treatment programs. (2) Soft tissue massage and joint ROM exercise as described in the early phase was implemented as well with intensity up to patient’s tolerance. Once the participants felt free of shoulder pain, the training was transferred to the open-chain exercise. (3) Progressive resistance exercise (PRE) training for the open-chain exercises was administered with free weights or thera-bands 1 set of 10 repetitions in one session (Fig. 2). In this phase, some of the participants started to receive RT, which can interfere with TMJ and oral function. (4) To maintain TMJ flexibility, conventional therapy for TMJ was maintained in these subjects. On top of that, oral function training was implemented as well. (5) Tongue and lip mobility and coordination training, and instructions on food intake were included. (6) In cases where a fibular osteoseptocutaneous flap was performed, the big toe and ankle joint were stretched in this phase to increase joint ROM and soft tissue flexibility. The training programs were evaluated and adjusted as needed during every visit. Each treatment session lasted for an average of 50 min. Participants were also instructed to perform individual home-programs on an hourly basis when awake at home. Compliance to home-programs was checked having the patients keep diaries regarding their home endeavors.

Figure 2 Progressive resistance exercise (PRE) training.

(A) Shoulder diagonal flexion with thera-band. (B) Shoulder horizontal abduction and scapulae retraction with thera-band. (C) Shoulder flexion and abduction with free weights.

Late phase

The late phase started more than three months post-operation. The goal in this phase was to recover the residual functions as much as possible. The interventions included were: (1) Pain and scar management with intensity up to patient’s tolerance and (2) shoulder PRE programs were implemented as in the previous phase, i.e., 10 repetitions each session. (3) If microstomia was caused by the intra-oral scar contracture after RT, a microstomia splint was prescribed and intra-oral scar massage was applied. Treatments to improve oral, shoulder joint and physical functioning were performed and progressed according to individual performance. The participants attended the rehabilitation center once a week. Each treatment session lasted for 1 h during this phase. In addition to a weekly visit to the rehabilitation center, patients were asked to perform their individual home programs with hourly exercise of the TMJ and shoulder, and were also encouraged to perform physical exercise (i.e., scar massage, mouth opening exercise, shoulder-neck ROM and PRE, et al.) at least three to five times a week.

All the intervention program is based on a variety of clinical trials compiled in the review article by Guru, Manoor & Supe (2012).

All the treatments were provided by three certified therapists with an average of more than 10 years of clinical experience.

Baseline data included demographic information and outcome variables taken during the 1st visit after operation. Data at one, three and six months after the operation were also included. Furthermore, return-to-work status of all subjects employed at baseline were followed at 12-month post operation through telephone interview to confirm if they are back to work.

Outcome measurements

Demographic data including gender, age, marital status, education level, disease staging, oncology and reconstructive treatment were collected after informed consent was given. Outcome measurements included shoulder outcomes, oral health, health-related QoL and physical functions. Shoulder outcomes were evaluated according to shoulder joint ROM and muscle strength (MMT), the visual analog scale (VAS) for the assessment of shoulder pain, and the Disabilities of the Arm, Shoulder and Hand (DASH) Outcome Measure for upper extremity function. Oral health was evaluated using maximum mouth opening (MMO) and diet status, which was divided into nasogastric (NG) tube feeding, liquid diet, soft diet or normal diet. Health-related QoL was evaluated using the European Organization for Research and Treatment of Cancer Quality of Life Questionnaire Core 30 (EORTC QLQ-C30) and Quality of Life Questionnaire Core Head and Neck 35 (QLQ-HN35), and using physical function evaluations. Six-minute walking distance (6MWD) and the timed up and go (TUG) tests were employed to evaluate physical function.

MMT and ROM

The strength of the rotator cuff, pectoralis muscle, deltoid, trapezius, serratus anterior, rhomboid and latissmus dorsi were evaluated according to the numerical grading system of Medical Research Council (MRC) scale. A score of 4 is the acknowledged cut-off score that demonstrates the muscle’s ability to hold testing position against gravity and moderate resistance (Mark, 2017). Shoulder joint ROM was evaluated in flexion, abduction, external rotation, and internal rotation with a two-arm goniometer under standard procedures. The evaluation was terminated when any contraindications in the prone position were indicated such as breathing difficulty. Interincisor distance (IID) was collected in millimeters by a Willis gauge, in the case of healthy teeth. Maximal mouth opening (MMO) was measured by measuring the difference between the distances between the nose tip to mandible during mouth opening and mouth closing.

DASH outcome measure

The DASH outcome measure is a self-report questionnaire and a clinical tool to measure physical function and symptoms of the upper extremity, which studies have found useful for patients after neck dissection (Carr, Bowyer & Cox, 2009; Chan et al., 2015). The disability/symptom section contains 30 items scored from 1 to 5 indicating “no difficulty” to “unable” to perform the task. At least 27 items must be completed, scores are consequently transformed into a total scale of 1–100. Higher scores indicate greater disability. A minimum clinically important difference (MCID) was used to assess the smallest perceived important change (Cranney et al., 2001). A change score exceeding 15 points is the most accurate change score for discriminating between improved and unimproved state (Beaton et al., 2001).

EORTC QLQ-C30 and QLQ-HN35

The EORTC QLQ-C30 is a questionnaire developed to evaluate the general quality of life of cancer patients. The QLQ-H&N35 is one of the disease-specific module supplements for head and neck cancer. The QLQ-H&N35, in conjunction with the QLQ-C30, is considered a reliable and valid assessment of the quality of life among patients with head and neck cancer in various different countries (Bjordal et al., 2000; Chie et al., 2010; Scherman, Simonton & Adams, 2000). The EORTC QLQ-C30 contains 30 questions and is divided into a global health scale, five functional (physical, role, cognitive, emotional, and social) scales, and nine symptom (fatigue, pain, nausea/vomiting, dyspnea, insomnia, appetite loss, constipation, diarrhea, and financial difficulties) scales. The EORTC QLQ-H&N35 contains 35 questions and is divided into six symptom scales (pain, swallowing, senses (taste/smell), speech, social eating, and social contact), and seven single items (impaired sexuality, teeth problems, mouth opening, dry mouth, sticky saliva, coughing, and feeling ill). Every scale is transformed into a score ranging from 0 to 100. A higher score on the functional scale or global health scale represents a higher level of functioning or quality of life. In contrast, a higher score on the symptom scale or single item scale reflects a worse symptom or problem.

Physical functions

The 6-minute walking distance (6MWD) and the timed up and go (TUG) tests were employed to evaluate physical capacity and lower extremity function. 6MWD is a tool used to evaluate functional exercise capacity of cancer patients and has been suggested to be as valid and reliable in this context as it is in evaluating cardiopulmonary patients (Schmidt et al., 2013). The test was performed in 6 min, and the patients were asked to walk to the end and back on a 20 m walkway as fast as they can, without overexertion. The total distance was recorded in millimeters. The TUG test is a commonly used tool that is evidently based to predict functional mobility and stability in less healthy, lower-functioning adults (Schoene et al., 2013). It measures the time in seconds to stand up from a sitting position on a chair, walk 3 m, turn, walk back, and to sit back down on the chair.

All the evaluations were conducted by a different group of two certified therapists blinded to the study, with an average 12.5 years of clinical experience.

Statistical analysis

SPSS 20.0 was used in the data analysis with a maximum significance level set at 0.05. Independent t-tests were used for the analysis of continuous variables. For the categorical data, chi-squared tests or Fisher’s exact tests were used to analyze differences in the independent variables of gender, marital status, educational level, breadwinner, vocation, TNM stage, neck dissection, radiation therapy, diet and donor site, while independent t-tests were applied to test for differences in the dependent variables of age, 6MWD and TUG. The Generalized Estimating Equations (GEE) procedure was conducted to analyze repeated measures outcome variables over time. We used the GEE model which assumed unstructured working correlation matrix. GEE has the benefit of a robust estimator, and it can overcome the limitation of missing data and adjust for correlations between observations as well. Separate models were run for each outcome variable. Bilateral shoulders were evaluated separately when bilateral neck dissections had been performed. The data were excluded if the participants had shoulder problems before the operation. Pearson correlation coefficient was used to measure the degree of association between outcome variables. Kaplan–Meier estimates were used for the survival analysis to predict the rate of return-to-work. The log rank test was conducted to compare differences in the rate of return-to-work between the early stage and advanced stage groups.

Results

Study population

A total of 65 participants (60 male and five female, mean age 51.6 years) who underwent 80 neck dissections (15 subjects received bilateral neck dissections), were analyzed in this study after reconstructive microsurgery. Thirteen eligible subjects declined to enroll. The drop-out rate was 34%. The reason was decondition and fatigue caused by RT that decreased their willingness to participate in the intervention program. The CONSORT flow diagram is shown in Fig. 3. The demographic data are summarized in Table 1. A supplementary table (Table S1) is provided to present the demographic comparison between the drop-out and the analytic groups. To explore differences in the outcomes between the early stage and the advanced stage group, we compared them with respect to all variables of interest (using group model). Apart from the disease-related variables, there were no significant differences in the demographic data of the early staged group and the advanced group.

Figure 3 The CONSORT flow chart.

Table 1 Demographic and clinical characteristics of the study participants.

		Early stage (n = 27)	Advanced stage (n = 38)	p-value	
Age	50.7 ± 9.9	51.8 ± 10.6	.660	
Gender	F	3 (11%)	2 (5%)	.642	
	M	24 (89%)	36 (95%)		
Education	<9 years	10 (39%)	16 (42%)	.878	
	9–12 years	13 (46%)	18 (47%)		
	>12 years	4 (15%)	4 (11%)		
Marriage	Single	3 (11%)	4 (11%)	1.000	
	Married	21 (78%)	30 (79%)		
	Other	3 (11%)	4 (11%)		
Breadwinner	N	10 (37%)	10 (27%)	.394	
	Y	17 (63%)	28 (73%)		
Vocation	Retired/unemployed	1 (4%)	7 (18%)	.253	
	Self-employed	7 (26%)	8 (21%)		
	Professional	3 (11%)	2 (5%)		
	Administration	2 (7%)	1 (3%)		
	Service	6 (22%)	4 (11%)		
	Semi-skilled/skilleds worker	8 (30%)	16 (42%)		
ND	Right	10 (37%)	15 (40%)	.002	
	Left	16 (59%)	9 (24%)		
	Bilateral	1 (4%)	14 (37%)		
Area of tumor	Buccal	11 (41%)	9 (24%)	.714	
	Tongue	6 (22%)	11 (29%)		
	Mouth floor	2 (7%)	6 (16%)		
	Gum	4 (15%)	4 (11%)		
	Pharyngeal wall	0 (0%)	2 (5%)		
	Lip	1 (4%)	2 (5%)		
	Retromolar tumor	2 (7%)	2 (5%)		
	Gingival	0 (0%)	1 (3%)		
	Hard palate	0 (0%)	1 (3%)		
	Soft palate	1 (4%)	0 (0%)		
T stage	1	8 (30%)	0 (0%)	<.001	
	2	19 (70%)	11 (29%)		
	3	0 (0%)	7 (18%)		
	4	0 (0%)	20 (53%)		
N stage	0	27 (100%)	10 (26%)	<.001	
	1	0 (0%)	8 (21%)		
	2	0 (0%)	19 (50%)		
	3	0 (0%)	1 (3%)		
M stage	0	27 (100%)	38 (100%)		
Radiation therapy	No	20 (74%)	8 (22%)	<.001	
	Yes	7 (26%)	30 (78%)		
Donor site	Anterolateral thigh flap	7 (26%)	8 (21%)	.089	
	Fibular osteoseptocutaneous flap	1 (4%)	9 (24%)		
	Other	19 (70%)	21 (55%)		

Shoulder outcomes

All participants had neck dissections, and none had a prior history of shoulder pain. With respect to shoulder ROM of the affected shoulder, only abduction showed a significant difference between six months and baseline or compared with the unaffected side during six months of the study (p < .001) (Tables 2 and 3). MMT of the middle trapezius, lower trapezius and rhomboid muscles were not performed at the baseline because patients were unable to comfortably maintain a prone posture. The muscle strength of the infraspinatus, subscapularis, teres muscles, pectoralis muscles and latissmus dorsi demonstrated normal strength at the baseline and exhibited no significant differences in the subsequent tests. Apart from the upper trapezius and levator scapular which gained stability of normal strength (4.9 ± 0.3) one month post operation (Table 2), the anterior and middle deltoid, trapezius, serratus anterior, rhomboid all showed significant differences in strength between the 6-month assessment compared to baseline, one month and three months, individually (p < .001) (Table 2). At one-month evaluation, only middle and lower trapezius scored less than 4 on the MMT scale. Other muscles attained sufficient strength to overcome some resistance and gravity. Comparison of the ratio of the shoulder outcomes of the neck dissection side and the sound side indicates that the shoulder ROM and muscle strength reached almost equivalent level to the sound side except in the middle and lower trapezius at six months of the study (Table 3).

Table 2 Progression of ROM of shoulder joint and of MMT of scapular stabilizers.

	Baseline	1 month	3 months	6 months	p-value for trend	
	Mean	SD	Mean	SD	Mean	SD	Mean	SD		
ROM (°)										
Flexion	152.1	20.7	151.3	22.1	155.3	21.3	158.9	14.2	.058	
Abduction	146.3	29.1	154.8	24.8	161.4	14.4	165.0	16.5	<.001	
IR	74.5	10.3	77.1	9.0	72.0	8.7	70.8	11.3	.108	
ER	79.7	14.8	82.3	10.2	81.3	10.0	82.2	11.2	.337	
MMT										
MT			4.2	0.9	4.4	0.7	4.8	0.4	<.001	
MLT			2.7	1.0	2.8	1.1	3.4	1.3	.001	
RMM			4.4	0.8	4.7	0.5	4.9	0.2	<.001	
SA	4.2	0.9	4.5	0.6	4.6	0.7	4.9	0.4	<.001	
LUT	4.6	0.8	4.9	0.3	4.9	0.3	5.0	0.2	<.001	
AD	4.4	0.7	4.7	0.5	4.8	0.6	4.9	0.3	<.001	
MD	4.2	0.9	4.6	0.7	4.6	0.8	4.9	0.4	<.001	
Notes.

ROM range of motion

IR internal rotation

ER external rotation

MMT manual muscle testing

MT middle trapezius

MLT middle and lower trapezius

RMM rhomboid major and minor

SA serratus anterior

LUT upper trapezius and levator scapular

AD anterior deltoid and coracobrachialis

MD middle deltoid and supraspinatus

Table 3 Comparison of the ratio of the shoulder joint ROM and MMT between the neck-dissection side and the sound side.

	Baseline	1 month	3 months	6 months	p-value for trend	
	Mean	SD	Mean	SD	Mean	SD	Mean	SD		
ROM										
Flexion	.93	.13	.92	.15	.95	.13	.97	.05	.027	
Abduction	.87	.17	.88	.16	.93	.06	.96	.05	<.001	
IR	1.06	.21	1.06	.12	1.02	.14	1.02	.13	.101	
ER	.97	.11	.99	.06	.96	.08	.98	.07	.467	
MMT										
MT			.93	.15	.94	.10	1.00	.16	.012	
MLT			.69	.21	.68	.22	.76	.23	.005	
RMM			.97	.08	.99	.03	1.00	0.00	.025	
SA	.93	.17	.90	.14	.92	.12	.98	.07	.240	
LUT	.94	.15	.98	.06	.99	.04	1.00	0.00	.026	
AD	.92	.13	.94	.12	.96	.10	.98	.06	.003	
MD	.90	.14	.92	.14	.95	.12	.98	.09	<.001	
Notes.

ROM range of motion

IR internal rotation

ER external rotation MMT manual muscle testing

MT middle trapezius

MLT middle and lower trapezius

RMM rhomboid major and minor

SA serratus anterior

LUT upper trapezius and levator scapular

AD anterior deltoid and coracobrachialis

MD middle deltoid and supraspinatus

The average shoulder pain score on the VAS was 1.81 ± 2.28 at the baseline, and 1.09 ± 1.89 at six months. VAS scores showed no significant difference between six month and baseline. However, the pain scale measured in the self-reported quality-of-life questionnaire EORTC QLQ-C30 showed a significantly higher score at the baseline (44.9 ± 26.0) (p < .001) compared with subsequent evaluations (Table 4). The mean DASH score at the baseline was 34.4 ± 24.0 and significantly dropped to 17.4 ± 16.2 at 1 month (p < .001), which was considered a minimum clinically important difference (MCID).

Table 4 Overview of the EORTC QLQ-C30 scores in the whole model and group model.

		Overall model	Group model	
				The early stage	The advanced stage	
		Mean ± SD	Estimate (95% CI)	Mean ± SD	Estimate (95% CI)	Mean ± SD	Estimate (95% CI)	
Global quality of life	Baseline	34.66 ± 28.88	−28.57*(−36.5, −20.6)	33.33 ± 27.74	−34.18*(−45.9, −22.4)	35.65 ± 30.05	−22.07*(−32.4, −11.8)	
	1m	54.94 ± 20.32	−7.19*(−13.4, −1.0)	56.06 ± 20.60	−10.58*(−18.5, −2.7)	54.17 ± 20.41	−2.69 (−11.4, 6.0)	
	3m	56.94 ± 19.75	−6.25*(−11.4, −1.1)	64.58 ± 20.22	−3.23 (−9.8, 3.3)	50.40 ± 17.09	−7.27*(−14.4, −0.2)	
	6m	62.33 ± 22.09	ref.	67.97 ± 26.46	ref.	55.93 ± 14.05	ref.	
Physical function	Baseline	62.97 ± 24.44	−21.37*(−27.3, −15.4)	68.89 ± 23.32	−20.01*(−30.1, −9.9)	58.77 ± 24.65	−21.98*(−28.5, −15.5)	
	1m	78.15 ± 19.44	−5.75*(−10.1, −1.4)	80.00 ± 16.84	−8.27*(−13.5, −3.0)	76.88 ± 21.22	−3.58 (−9.6, 2.4)	
	3m	81.56 ± 17.33	−2.27 (−5.8, 1.3)	85.79 ± 15.38	−2.38 (−6.6, 1.8)	77.94 ± 18.34	−2.11 (−7.6, 3.4)	
	6m	84.48 ± 15.79	ref.	86.99 ± 18.97	ref.	81.63 ± 11.13	ref.	
Role function	Baseline	45.38 ± 37.79	−32.18*(−43.4, −21.0)	48.15 ± 36.20	−33.81*(−48.2, −19.4)	43.42 ± 39.24	−30.04*(−46.8, −13.3)	
	1m	67.90 ± 26.27	−9.60 (−19.9, 0.7)	69.70 ± 21.60	−11.52 (−24.2, 1.1)	66.67 ± 29.33	−7.09 (−22.7, 8.5)	
	3m	73.18 ± 24.17	−3.47 (−10.3, 3.4)	77.43 ± 19.46	−3.43 (−12.2, 5.4)	69.54 ± 27.40	−2.97 (−13.2, 7.3)	
	6m	74.13 ± 26.54	ref.	77.94 ± 31.86	ref.	69.81 ± 19.04	ref.	
Emotional function	Baseline	68.10 ± 23.83	−7.98*(−13.6, −2.4)	66.36 ± 25.99	−9.72*(−19.2, −0.2)	69.37 ± 22.40	−6.39*(−12.4, −0.4)	
	1m	77.78 ± 18.53	2.24 (−2.3, 6.8)	76.52 ± 19.69	1.55 (−5.9, 9.0)	78.65 ± 17.95	3.05 (−1.8, 7.9)	
	3m	75.45 ± 20.38	−0.67 (−4.2, 2.9)	75.58 ± 20.31	0.10 (−4.7, 4.9)	75.35 ± 20.81	−1.13 (−6.1, 3.8)	
	6m	75.17 ± 21.06	ref.	76.14 ± 25.18	ref.	74.07 ± 15.96	ref.	
Cognitive function	Baseline	74.22 ± 21.81	−5.75 (−12.2, 0.7)	72.22 ± 21.68	−9.57 (−19.9, 0.7)	75.68 ± 22.08	−1.83 (−8.8, 5.1)	
	1m	81.48 ± 17.03	2.17 (−3.8, 8.1)	79.55 ± 19.88	−0.73 (−10.6, 9.2)	82.81 ± 14.96	5.47 (−1.1, 12.1)	
	3m	80.34 ± 19.05	0.14 (−3.7, 4.0)	81.25 ± 19.37	−0.18 (−4.9, 4.5)	79.56 ± 19.08	1.28 (−4.4, 7.0)	
	6m	78.65 ± 19.96	ref.	81.54 ± 22.48	ref.	75.37 ± 16.83	ref.	
Social function	Baseline	48.96 ± 28.77	−20.08*(−28.6, −11.6)	53.70 ± 27.48	−20.36*(−32.8, −8.0)	45.50 ± 29.57	−18.84*(−30.2, −7.5)	
	1m	64.20 ± 22.06	−4.73 (−11.7, 2.2)	65.15 ± 21.15	−8.33 (−19.3, 2.6)	63.54 ± 22.97	−1.11 (−9.3, 7.1)	
	3m	69.71 ± 23.62	0.40 (−5.0, 5.8)	74.19 ± 19.94	0.67 (−6.3, 7.6)	65.87 ± 26.11	0.77 (−7.3, 8.8)	
	6m	69.79 ± 22.74	ref.	73.86 ± 25.14	ref.	65.19 ± 19.50	ref.	
Fatigue	Baseline	42.48 ± 25.24	18.78*(11.1, 26.4)	43.21 ± 24.33	22.45*(10.4, 34.5)	41.96 ± 26.18	14.96*(5.8, 24.1)	
	1m	30.04 ± 15.21	6.20*(0.2, 12.2)	31.31 ± 16.31	10.32*(1.9, 18.7)	29.17 ± 14.60	2.06 (−5.8, 9.9)	
	3m	28.45 ± 18.23	5.11*(0.5, 9.7)	25.31 ± 20.42	4.23 (−0.2, 8.6)	31.15 ± 15.99	5.13 (−2.8, 13.1)	
	6m	24.48 ± 21.18	ref.	21.57 ± 25.26	ref.	27.78 ± 15.57	ref.	
Nausea/vomiting	Baseline	13.85 ± 19.22	8.04*(2.4, 13.7)	14.81 ± 22.33	9.23*(0.7, 17.8)	13.16 ± 16.96	7.96*(0.3, 15.7)	
	1m	8.33 ± 18.24	2.38 (−2.5, 7.3)	2.27 ± 7.79	−3.64 (−9.1, 1.8)	12.50 ± 22.00	7.32*(0.3, 14.3)	
	3m	10.31 ± 14.01	4.92*(0.5, 9.3)	5.32 ± 12.09	0.09 (−6.2, 6.4)	14.58 ± 14.32	9.72*(4.3, 15.2)	
	6m	5.64 ± 11.41	ref.	4.74 ± 12.02	ref.	6.67 ± 11.00	ref.	
Pain	Baseline	44.87 ± 26.01	22.48*(14.0, 31.0)	45.68 ± 20.46	26.15*(13.9, 38.4)	44.30 ± 29.58	18.55*(7.5, 29.6)	
	1m	24.38 ± 19.88	1.92 (−4.9, 8.7)	27.27 ± 18.93	7.89 (−0.6, 16.3)	22.40 ± 20.57	−3.58 (−12.6, 5.5)	
	3m	28.21 ± 20.03	5.63*(0.2, 11.1)	25.12 ± 20.84	4.66 (−1.8, 11.1)	30.85 ± 19.28	5.62 (−3.0, 14.2)	
	6m	22.05 ± 21.40	ref.	19.93 ± 24.93	ref.	24.44 ± 17.10	ref.	
Dyspnea	Baseline	23.44 ± 28.28	12.73*(4.7, 20.8)	20.99 ± 20.98	13.55*(3.4, 23.6)	25.23 ± 32.78	13.76*(1.3, 26.2)	
	1m	8.64 ± 17.36	−1.73 (−6.5, 3.1)	7.58 ± 17.61	−0.10 (−3.0, 2.8)	9.38 ± 17.42	−1.70 (−9.8, 6.4)	
	3m	13.46 ± 14.98	2.70 (−2.5, 7.9)	8.80 ± 13.50	0.70 (−5.5, 6.9)	17.46 ± 15.25	5.61 (−1.9, 13.1)	
	6m	10.94 ± 17.25	ref.	7.84 ± 18.74	ref.	14.44 ± 15.26	ref.	
Insomnia	Baseline	48.21 ± 32.82	24.28*(14.5, 34.1)	46.91 ± 33.66	27.14*(12.9, 41.3)	49.12 ± 32.64	20.26*(6.4, 34.1)	
	1m	29.63 ± 27.98	4.39 (−3.7, 12.5)	36.36 ± 30.70	14.98*(4.9, 25.0)	25.00 ± 25.40	−4.98 (−17.4, 7.4)	
	3m	28.21 ± 23.75	4.53 (−1.6, 10.7)	25.93 ± 21.34	6.84*(0.2, 13.5)	30.16 ± 25.86	1.34 (−8.8, 11.5)	
	6m	25.35 ± 27.83	ref.	18.95 ± 23.41	ref.	32.59 ± 31.35	ref.	
Loss of appetite	Baseline	25.13 ± 30.64	10.00*(0.2, 19.8)	28.40 ± 32.95	20.59*(8.7, 32.4)	22.81 ± 29.11	−0.32 (−15.5, 14.8)	
	1m	13.58 ± 21.98	−1.91 (−9.4, 5.5)	10.61 ± 18.93	2.36 (−3.6, 8.4)	15.63 ± 23.92	−7.88 (−21.6, 5.8)	
	3m	20.09 ± 19.96	5.26 (−1.7, 12.2)	14.35 ± 17.49	6.68*(0.4, 13.0)	25.00 ± 20.92	2.37 (−10.5, 15.2)	
	6m	15.63 ± 23.59	ref.	8.17 ± 14.84	ref.	24.07 ± 28.92	ref.	
Constipation	Baseline	20.00 ± 26.87	1.72 (−6.1, 9.6)	18.52 ± 28.24	3.03 (−4.3, 10.3)	21.05 ± 26.19	−0.02 (−14.3, 14.3)	
	1m	16.05 ± 23.11	−3.31 (−10.2, 3.6)	13.64 ± 24.47	−2.43 (−7.8, 3.0)	17.71 ± 22.38	−4.47 (−17.7, 8.8)	
	3m	19.23 ± 21.88	1.73 (−3.3, 6.8)	14.12 ± 19.45	1.19 (−3.8, 6.2)	23.61 ± 23.21	2.16 (−7.2, 11.6)	
	6m	17.53 ± 25.05	ref.	14.38 ± 22.74	ref.	21.11 ± 27.79	ref.	
Diarrhea	Baseline	22.40 ± 23.80	7.65*(0.6, 14.7)	27.16 ± 26.21	15.58*(6.2, 25.0)	18.92 ± 21.57	−0.61 (−10.6, 9.4)	
	1m	17.90 ± 19.11	3.55 (−2.4, 9.5)	19.70 ± 19.68	8.67*(1.0, 16.3)	16.67 ± 18.93	−2.64 (−10.7, 5.4)	
	3m	11.75 ± 15.39	−2.83 (−7.8, 2.2)	12.27 ± 17.34	0.57 (−5.4, 6.6)	11.31 ± 13.82	−7.72*(−15.2, −0.2)	
	6m	13.89 ± 15.90	ref.	8.50 ± 14.45	ref.	20.00 ± 15.69	ref.	
Financial problems	Baseline	42.71 ± 29.97	11.54*(1.0, 22.0)	35.80 ± 29.13	14.73*(1.1, 28.4)	47.75 ± 29.96	8.10 (−7.9, 24.1)	
	1m	32.72 ± 30.02	0.64 (−8.9, 10.2)	21.21 ± 24.22	−0.38 (−9.9, 9.2)	40.63 ± 31.38	0.09 (−16.3, 16.5)	
	3m	25.21 ± 26.00	−6.03 (−12.4, 0.4)	18.06 ± 23.81	−3.60 (−11.5, 4.3)	31.35 ± 26.63	−8.72 (−19.1, 1.7)	
	6m	30.56 ± 31.80	ref.	23.53 ± 31.21	ref.	38.52 ± 31.60	ref.	
Notes.

* p < 0.05.

ref, reference group (the base for comparison).

Oral health outcomes

At the baseline, a nasogastric (NG) tube was placed for all participants. One month after the operation, 18% persisted with NG tube feeding, 14% returned to normal diet, 28% and 40% started to get nutrition with a liquid and a soft diet, respectively. At six months, all NG tubes had been removed, and 53% returned to normal diet. 16% and 31% participants were able to orally intake a liquid and a soft diet.

IID and MMO had high positive correlation (r = .883). The mean IID was 22.5 ± 9.2 millimeters and significantly progressed to 31.8 ± 11.8 millimeters (p < .001) in the overall model. The mean MMO was 22.6 ± 2.1 millimeters (baseline) and 31.8 ± 10.1 millimeters (6 months), which was also a statistically significant difference (p < .001). In the group model, IID and MMO for both early stage and advanced stage groups significantly increased at 6 months compared to baseline, one and three months, individually as well. In the early stage group, IID increased from 21.8 ± 7.9 (baseline) to 34.0 ± 13.7 (six months) (p < .001), and it increased from 23.1 ± 10.1 (baseline) to 29.1 ± 8.6 (six months) (p = .010) in the advanced stage group.

Health related quality of life and physical functions

Table 4 illustrates the results of the EORTC QLQ-C30 scales. After six months of intensive physical therapy, health-related quality of life was significantly different (p < .05) from the baseline on the global health scale and all functional scales, except cognitive function. The symptom scales of fatigue, nausea/vomiting, pain, dyspnea and insomnia showed significant differences between the baseline and six-month scores or between three-month and 6-month scores (p < .05) for the overall and group models. However, no significant difference was found on the scales of constipation over time throughout the six months. While the scales of loss of appetite, diarrhea, and financial problems showed significantly high scores (p < .05) at baseline compared with 6-month scores in the early stage group, it maintained at similar level over time throughout the 6 months in the advanced stage group.

Table 5 illustrates the results from the EORTC QLQ-H&N35 scales. All other symptom scales showed a significant difference (p < .05) between the six-month test scores and the baseline scores in overall model and group model except on the scale on senses (tastes, smell) which showed no significant difference over six months in the early stage group. Oral pain, swallowing, and speech scores on the three-month test were significantly different (p < .05) compared with those on the six-month test in the advanced group. The single item scales of opening mouth, dry mouth, sticky saliva and coughing showed significant differences between the baseline and six-month scores, and between 3-month and 6-month scores (p < .05) except for the score on teeth and sexuality, which maintained at the same level over the 6 months in the early stage group and the advanced group.

Table 5 Overview of the EORTC QLQ-H&N35 scores in the whole model and group model.

		Overall model	Group model	
				The early stage	The advanced stage	
		Mean ± SD	Estimate (95% CI)	Mean ± SD	Estimate (95% CI)	Mean ± SD	Estimate (95% CI)	
Oral pain	Baseline	41.28 ± 27.03	19.90*(12.9, 26.9)	44.44 ± 24.02	24.18*(14.6, 33.8)	38.96 ± 29.14	15.80*(6.6, 25.0)	
	1m	23.15 ± 14.27	1.80 (−3.0, 6.6)	25.76 ± 13.83	5.75 (−1.3, 12.8)	21.35 ± 14.50	−1.90 (−7.0, 3.2)	
	3m	28.39 ± 18.29	6.43*(1.9, 11.0)	22.28 ± 16.72	1.72 (−4.7, 8.2)	33.63 ± 18.22	9.98*(4.8, 15.2)	
	6m	19.71 ± 15.54	ref.	18.06 ± 19.02	ref.	21.73 ± 10.20	ref.	
Swallowing	Baseline	56.21 ± 28.19	32.34*(25.4, 39.3)	46.60 ± 23.91	28.12*(18.2, 38.0)	63.21 ± 29.29	34.95*(25.7, 44.2)	
	1m	27.47 ± 19.67	4.19 (−1.0, 9.4)	26.14 ± 17.12	8.45*(2.5, 14.4)	28.39 ± 21.47	0.67 (−6.4, 7.8)	
	3m	32.48 ± 22.87	8.21*(3.1, 13.3)	22.11 ± 17.09	2.43 (−2.6, 7.5)	41.37 ± 23.69	12.91*(5.0, 20.8)	
	6m	24.60 ± 18.54	ref.	20.18 ± 21.15	ref.	29.96 ± 13.64	ref.	
Senses	Baseline	27.34 ± 31.90	9.02*(0.7, 17.4)	22.84 ± 28.17	9.28 (−2.5, 21.0)	30.63 ± 34.36	8.56 (−3.9, 21.0)	
	1m	13.58 ± 19.17	−4.44 (−11.4, 2.5)	11.36 ± 19.51	−2.86 (−11.5, 5.8)	15.10 ± 19.10	−5.58 (−17.0, 5.8)	
	3m	25.53 ± 21.61	6.97*(0.8, 13.2)	15.86 ± 15.44	2.47 (−4.6, 9.5)	33.83 ± 22.89	10.71*(0.4, 21.0)	
	6m	18.10 ± 23.97	ref.	13.07 ± 20.43	ref.	24.21 ± 27.18	ref.	
Speech	Baseline	52.43 ± 30.70	25.93*(17.1, 34.8)	48.56 ± 26.36	26.36*(13.2, 39.5)	55.26 ± 33.59	24.72*(13.0, 36.4)	
	1m	29.42 ± 20.83	4.18 (−2.3, 10.7)	31.31 ± 20.75	9.26*(0.7, 17.8)	28.13 ± 21.12	−0.20 (−9.0, 8.6)	
	3m	33.26 ± 25.06	6.21*(0.6, 11.8)	25.39 ± 17.42	2.62 (−4.8, 10.1)	40.01 ± 28.73	8.81*(1.0, 16.6)	
	6m	26.58 ± 24.30	ref.	22.66 ± 25.45	ref.	31.35 ± 22.81	ref.	
Social eating	Baseline	50.39 ± 32.71	14.08*(5.8, 22.4)	47.84 ± 32.08	15.97*(2.9, 29.1)	52.25 ± 33.49	11.87*(2.3, 21.4)	
	1m	33.33 ± 26.35	−2.33 (−8.5, 3.8)	35.23 ± 29.87	1.22 (−6.8, 9.2)	32.03 ± 24.05	−6.01 (−14.1, 2.1)	
	3m	37.34 ± 27.16	1.47 (−3.2, 6.1)	32.99 ± 24.83	−0.19 (−6.3, 5.9)	41.07 ± 28.93	2.26 (−4.3, 8.8)	
	6m	36.51 ± 24.37	ref.	33.50 ± 30.21	ref.	40.18 ± 14.85	ref.	
Social contact	Baseline	43.44 ± 29.07	14.93*(6.3, 23.5)	39.26 ± 24.34	14.95*(2.7, 27.2)	46.49 ± 32.07	14.62*(2.7, 26.5)	
	1m	28.40 ± 22.40	−0.68 (−7.2, 5.8)	30.00 ± 23.44	3.49 (−5.5, 12.5)	27.29 ± 21.97	−4.10 (−13.2, 5.0)	
	3m	30.67 ± 24.38	2.67 (−3.5, 8.8)	27.38 ± 21.67	3.11 (−4.4, 10.6)	33.49 ± 26.55	2.25 (−7.4, 11.9)	
	6m	28.61 ± 20.88	ref.	26.40 ± 22.38	ref.	30.72 ± 19.62	ref.	
Sexuality	Baseline	39.32 ± 35.68	12.24*(3.2, 21.3)	34.57 ± 27.71	12.26 (−0.5, 25.0)	42.79 ± 40.55	10.85 (−1.5, 23.2)	
	1m	29.94 ± 26.77	2.10 (−6.1, 10.3)	31.06 ± 26.38	6.24 (−2.3, 14.7)	29.17 ± 27.44	−2.52 (−15.4, 10.4)	
	3m	33.07 ± 29.03	5.77 (−0.2, 11.7)	28.01 ± 26.58	4.73 (−3.0, 12.5)	37.40 ± 30.79	5.51 (−3.6, 14.6)	
	6m	29.57 ± 29.65	ref.	24.02 ± 33.58	ref.	36.31 ± 23.48	ref.	
Teeth	Baseline	46.03 ± 36.13	5.86 (−5.8, 17.5)	41.98 ± 30.09	5.75 (−8.0, 19.5)	49.07 ± 40.23	5.33 (−13.5, 24.2)	
	1m	37.65 ± 32.41	−2.85 (−13.0, 7.3)	42.42 ± 31.17	5.86 (−5.6, 17.3)	34.38 ± 33.32	−9.75 (−26.2, 6.7)	
	3m	40.71 ± 26.83	−0.47 (−8.8, 7.9)	33.33 ± 24.08	−5.34 (−14.2, 3.6)	47.02 ± 27.86	3.22 (−11.2, 17.6)	
	6m	43.37 ± 33.62	ref.	40.52 ± 34.79	ref.	46.83 ± 33.10	ref.	
Opening mouth	Baseline	51.56 ± 34.60	20.97*(12.0, 29.9)	48.15 ± 32.47	25.05*(12.1, 38.0)	54.05 ± 36.30	14.51*(3.2, 25.8)	
	1m	37.04 ± 30.83	6.69 (−1.7, 15.1)	40.91 ± 32.42	18.96*(7.6, 30.3)	34.38 ± 29.92	−5.52 (−15.6, 4.5)	
	3m	36.43 ± 24.64	4.94 (−1.1, 10.9)	34.03 ± 23.07	9.05*(1.6, 16.5)	38.49 ± 26.14	−1.15 (−9.5, 7.3)	
	6m	32.08 ± 27.51	ref.	23.86 ± 26.99	ref.	42.06 ± 25.57	ref.	
Dry mouth	Baseline	42.71 ± 29.97	4.93 (−6.6, 16.4)	44.44 ± 30.66	18.64*(4.3, 33.0)	41.44 ± 29.82	−10.77 (−25.0, 3.5)	
	1m	31.48 ± 22.82	−6.51 (−17.4, 4.4)	37.88 ± 23.67	12.10 (−2.0, 26.2)	27.08 ± 21.48	−25.49*(−37.1, −13.9)	
	3m	39.85 ± 23.54	1.04 (−6.7, 8.8)	33.33 ± 21.17	6.14 (−2.7, 15.0)	45.44 ± 24.38	−7.35 (−18.8, 4.1)	
	6m	39.61 ± 32.10	ref.	28.43 ± 33.79	ref.	53.17 ± 24.72	ref.	
Sticky saliva	Baseline	50.52 ± 35.13	18.25*(7.0, 29.5)	49.38 ± 36.25	21.47*(6.7, 36.3)	51.35 ± 34.78	15.50 (−1.2, 32.2)	
	1m	30.86 ± 23.21	−1.24 (−11.5, 9.1)	27.27 ± 19.62	−0.49 (−14.1, 13.1)	33.33 ± 25.40	−2.33 (−17.6, 12.9)	
	3m	40.71 ± 27.65	8.32*(0.0, 16.6)	28.47 ± 24.37	−0.21 (−9.3, 8.9)	51.19 ± 26.29	15.50*(2.4, 28.6)	
	6m	31.72 ± 29.99	ref.	28.76 ± 32.65	ref.	35.32 ± 27.18	ref.	
Coughing	Baseline	46.88 ± 27.68	25.39*(17.7, 33.1)	39.51 ± 22.72	25.07*(13.8, 36.4)	52.25 ± 29.96	24.25*(14.7, 33.8)	
	1m	25.93 ± 24.80	4.58 (−2.3, 11.4)	21.21 ± 16.41	6.55 (−3.5, 16.6)	29.17 ± 29.02	1.73 (−5.8, 9.3)	
	3m	30.45 ± 21.09	9.38*(3.8, 15.0)	22.92 ± 15.40	8.38*(1.1, 15.7)	36.90 ± 23.34	9.45*(1.4, 17.5)	
	6m	19.18 ± 16.22	ref.	13.73 ± 16.91	ref.	25.79 ± 13.00	ref.	
Feeling ill	Baseline	57.81 ± 92.44	28.78*(6.9, 50.6)	71.60 ± 136.41	41.73 (−7.8, 91.3)	47.75 ± 35.61	21.61*(9.9, 33.4)	
	1m	26.54 ± 25.39	−1.45 (−8.6, 5.7)	31.82 ± 26.18	3.26 (−6.6, 13.2)	22.92 ± 24.59	−2.32 (−12.1, 7.4)	
	3m	34.62 ± 22.32	5.09 (−1.1, 11.3)	32.41 ± 20.96	2.12 (−5.4, 9.6)	36.51 ± 23.64	9.86*(0.9, 18.8)	
	6m	29.03 ± 20.47	ref.	29.08 ± 27.18	ref.	28.97 ± 7.60	ref.	
Notes.

* p < 0.05.

ref., reference group (the base for comparison).

At three-months and six-months, 6MWD was 339.0 ± 54.4m and 381.2 ± 69.9m, respectively (p < .001). TUG was 10.6 ± 2.4 s at 3 months and 8.4 ± 1.5 s at 6-months (p < .001). 6MWD on the 3-month and 6-month tests, respectively, was 355.5 ± 55.9m and 396.5 ± 73.5m (p = .029) in the early stage group, and 321.6 ± 48.2m and 363.1 ± 64.0m in the advanced stage group (p = .003). No significant difference was found between groups at 3 or 6 months on either measure.

Our analysis found a predicted return-to-work rate of 34.7% at three months, 63.0% at the six months and 80% at one year after surgery, respectively. The log rank test showed the time of return-to-work was significantly different between the early stage group and the advanced stage (p = .032). In the early stage group, it predicted that about 80% would return to work one year after the operation, but around 50% would return to work in the advanced stage group (Fig. 4). Further analysis showed 71.4% return-to-work rate in the retired or unemployed, 86.7% in the self-employed, 80% in the professional, 33.3% in the administration, 81.8% in the service and 87.5% in the semi-skilled or skilled according to vocational type.

Figure 4 Curve of the percentage of return-to-work patients at post-operative month in the early and advanced stage groups.

Discussion

In this single-arm pilot study we investigated early physical therapy interventions designed to improve active shoulder abduction and muscle strength surrounding the scapula. Shoulder functions including joint ROM, MMT, pain and DASH in the initial six- month post-operation were measured. In the early phase, the average shoulder ROM was limited to less than one-third the normal range, after which shoulder abduction had significantly improved at six-months post-surgery. In the model analyzed by the GEE procedure, the strength of the muscle group responsible for scapular stabilization, consisting of the trapezius, serratus anterior, and rhomboid, significantly improved during the six months. Furthermore, the deltoid and supraspinatus muscle also showed improvements in this study. Average shoulder pain measured by VAS did not exceed 2 at the baseline and the following six months. The DASH outcome measure reached MCID at the one-month evaluation, and maintained up to six months after the operation.

Ewing and Martin found shoulder problems after neck dissection with a clinical picture that included drooping of the shoulder, limited forward flexion, lateral abduction and rotation of the shoulder and reduced EMG activity (Ewing & Martin, 1952). EMG studies also showed that trapezius muscle activity decreased after neck dissection (Lima, Amar & Lehn, 2011; McGarvey et al., 2013a; Parikh et al., 2012). In our study, shoulder ROM and upper trapezius strength almost fully recovered during the early stage of physical therapy intervention, and scapular muscle strength showed continuous progress throughout the 6-month study period after the operation. PRE training has been found to improve shoulder function compared with standard intervention following neck dissection in randomized controlled trials (McNeely et al., 2004; McNeely et al., 2008). The timing of the physical therapy intervention in these previous studies was categorized into early and late stages. Early intervention seems to have some positive impact on this diagnostic group of patients and its change persisted for the entire six months of observation.

In previous studies, the target muscles included the upper trapezius ignoring the lower trapezius. An interesting finding of this study is that the upper trapezius gained normal level of strength soon after one month of intervention, but the middle and lower trapezius did not show a return to normal levels until after six months of intervention. In our model, the VAS and DASH scores showed mild disturbance at six months after surgery. This may be related to the impaired middle and lower trapezius, and needs further study in the future. In the reviewed articles, support of the beneficial effect of physical therapy for ANSD following neck dissection lacks convincing evidence (Bradley et al., 2011; Carvalho, Vital & Soares, 2012; McGarvey et al., 2011). In recent studies, PRE was introduced for the target muscles acting as scapular stabilizers (the upper trapezius, middle trapezius, rhomboid major, and serratus anterior muscle) rather than for the muscles surrounding the glenohumeral joint (Lima, Amar & Lehn, 2011; McGarvey et al., 2013a; McGarvey et al., 2013b). Improvements were observed in scapular muscles receiving PRE within six months after the operation. We suggest that lower trapezius muscle strengthening should be considered in the ANSD population.

TMJ ROM was measured by IID and MMO in this study. Our participants started to exercise their TMJ once the wound was stable, at an average of 8.3-days post-operation. Briefly speaking, our strategy was to promote early mobilization of the TMJ. The results showed a near 10 mm increase in TMJ ROM in six months. In the group model, the potential was more obvious in the early stage group. Early intervention to exercise the TMJ, may improve mouth opening. In the advanced stage group, the maximum mouth opening reached its highest at three months. This phenomenon may be related to RT. In our series, a higher rate of patients needed to receive RT among the advanced stage patients, which is consistent with other studies (Wetzels et al., 2014). We measured MMO by the method described by McCord & Grant (2000). This method can overcome the issue of missing teeth, and was highly correlated with the distance between incisors measured by a Willis gauge after our analysis.

We tested 6MWD to evaluate the prognosis of functional capacity and lower extremity function. Bellet, Adams & Morris (2012) systematically reviewed fifteen articles and concluded that 6MWD has strong evidence in support of its use to evaluate clinical change following cardiac rehabilitation 6MWD is significantly correlated with peak aerobic capacity (VO2peak). It is considered the gold standard outcome and is easily accessible in clinical practice (Schmidt et al., 2013). Our results showed that 6MWD significantly improved between the 3- and 6-month evaluations in both the overall and the group model. In Liu’s study, they performed the 6MWD test before and after 3 months of home-based exercise to evaluate lower extremity function after fibula osteocutaneous flap reconstruction for mandibular defects in oral cancer patients (Liu et al., 2013). Their results showed no significant difference in the walking distance. The exercise intervention was almost 1 year after surgery. In our series, the donor sites were harvested mostly from the lower extremity. The total walking distance and timed up and go (TUG) scores increased significantly in the first three months in both the overall and group model. Improved functional capacity and mobility may have been a factor contributing to early return-to-work. Return to work in many cancer survivors is a realistic outcome. Further study of the effect of rehabilitation on functional restoration and quality of life for oral cancer survivors is needed. In this study, our results suggested that early intervention and good compliance might help 80% of oral cancer survivors return to work within one year post-operation. We found a significant difference in the group model indicating that the cohort of early stage survivors returned faster and had a higher rate of return than the cohort of advanced stage survivors. In addition, return-to-work rate differs in different vocational types, with a higher rate of return in the skilled or semi-skilled and self-employed. In this study, samples are small in number in varied vocational types, further statistical analysis would convey little information. However, this would be an issue worth further investigation.

In this six-month goal-oriented rehabilitation program for the population of oral cancer survivors, the health-related QoL were observed to improve. In the domains measured by the EORTC QLQ-C30 scales and EORTC QLQ-H&N35 scales, the global health and other functional scales, except for cognitive function, improved significantly in the first month. Global health and physical function continued to improve throughout the 6 months. In our comparison of the early stage group and the advanced stage group, we found that global health and physical function deteriorated at the three-month evaluation in the advanced stage survivors. In some of the symptom and single item scales the same trend was observed, with the participant self-ratings being the worst at three months. These items included oral pain, swallowing, senses, speech, sticky saliva, feeling ill, and global health. As these are common side effects of radiation, they received low self-ratings at post-operative month 3 may be explained by the onset of RT. This finding is consistent with other studies (Ch’ng et al., 2014; Tribius et al., 2015). We discovered that the progression of some self-rated scales might be different in the group model. For example, global health reached a relatively stable score by the third month in the early stage group, whereas the score decreased at the three-month evaluation compared to baseline and recovered to the level measured at post-operative month 1 in the advanced stage group by the six-month evaluation. Self-rated health is a cognitive process, which is affected by internal and external factors (Huisman & Deeg, 2010; Jylha, 2009). We evaluated the outcomes of an early- and integrated-intervention program focused primarily on oral cavity function and secondarily on issues that result from the oncological and reconstructive surgery, and found significant improvement on several items within the six month study period.

The limitation of this study is the rather high loss rate up to 34%. In addition, follow-up lasted for only six months after surgery and the limited number of subjects recruited in this study. The shoulder pain and DASH scores still indicated mild impairment at the six-month evaluation. As a single-arm pilot study, tracking outcomes of subjects who had received oral cancer reconstruction surgery with early intervention rehabilitation program is our main objective. The results herein convey a possible application of early intervention to this diagnostic group of patients. The effect of early PRE intervention to the target muscles would be a worthwhile focus of future studies in this population. In addition, substantial outcomes were evaluated on limited number of subjects recruited in this study which might lead to a false-positive analytic results. An inclusion of a control group may warrant a better estimate of the effect of an early intervention to this group of patients as well.

Conclusion

Our results showed that shoulder joint range of motion and muscle strength progressively improved during the six months of observation, as did oral function and global health. Though subjects in this study underwent a comprehensive intervention program lasting only six months, all showed continued improvements in oral, upper extremity and lower extremity functions, as well as recovery of social roles (return to work). We suggest an early and integrated intervention, and a follow-up of at least six months following reconstructive surgery for future studies and clinical trials for oral cancer survivors; however, with discretion toward patients receiving radiation therapy.

Supplemental Information

Supplemental Information 1 TREND Statement Checklist

Non randomized trail checklist.

Click here for additional data file.

Supplemental Information 2 Raw data 1

Click here for additional data file.

Supplemental Information 3 Raw data 2

Click here for additional data file.

Supplemental Information 4 Demographic and clinical characteristics of the total participants

The demographic comparison between the drop-out and the analytic groups.

Click here for additional data file.

Supplemental Information 5 Clinical trial protocol

Click here for additional data file.

The authors sincerely thank the clinical staffs and participants in their contribution with regards to data collection. We also thank the Center for Big Data Analytics and Statistics for statistical support (Project No. CLRPG3D0044).

Additional Information and Declarations

Competing Interests

Author Contributions

Clinical Trial Ethics

Data Availability

Clinical Trial Registration

The authors declare there are no competing interests.

Yueh-Hsia Chen conceived and designed the experiments, performed the experiments, analyzed the data, prepared figures and/or tables.

Wei-An Liang, Siang-Lan Guo and Shwu-Huei Lien performed the experiments.

Chung-Yin Hsu edited.

Hsiao-Jung Tseng analyzed the data, prepared figures and/or tables.

Yuan-Hung Chao conceived and designed the experiments, analyzed the data, contributed reagents/materials/analysis tools, prepared figures and/or tables, authored or reviewed drafts of the paper.

The following information was supplied relating to ethical approvals (i.e., approving body and any reference numbers):

This study was approved by the Chang Gung Medical Foundation Institutional Review Board (Approval No: 103-5164B, 104-2300C, 104-8154C).

The following information was supplied regarding data availability:

The raw data is included in the Supplemental Files, and in the tables in the manuscript.

The following information was supplied regarding Clinical Trial registration:

NCT03206242.

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
