# Peer review of "Functional outcomes and quality of life after a 6-month early intervention program for oral cancer survivors: a single-arm clinical trial"

_PeerJ, doi:10.7717/peerj.4419_

## Round 0.1 · original submission · Major Revisions

· Academic Editor

Major Revisions

The article presents a detailed analysis of this “preliminary pilot study” – the intervention appears to be very considered and the initial results are promising in this group of patients. There are a number of issues the reviewers have noted which need to be addressed. There are a few minor points I’ve added below.

One main issue is the over interpretation of the results in the context of your design – a pilot study, even as detailed as this, should lay a foundation for subsequent research. Given the lack of control arm it is not appropriate to conclude that this intervention “is necessary” for these patients (lines 42, 422). It would be reasonable to conclude that it is worthy of further research.

Minor points not otherwise mentioned:

- P-values can not equal 0 (by definition). They should always be reported to 2 significant figures (or < 0.001). Similarly, p-values should be reported (rather than indicated as stars) where conducted.

- It is not stated exactly how many statistical tests have been performed? This should be stated along with a summary of how many were “significant” by your 0.05 criteria. A limitation of multiple significance testing is the increased rate of false-positives and this should be acknowledged.

- The cumulative incidence plot should indicate points where patients were censored and or the “number at risk” at each time point. The log-rank test and key summary statistics should be added to the figure.

Reviewer 1 ·

Basic reporting

Overall the manuscript is written using appropriate scientific language in the English language. There is a sufficient list of supportive references. The standard sections and subheadings are used throughout the manuscript as suggested by the journal.
However, the manuscript lacks clarity throughout, and is repeatedly ambiguous. Listed below are several suggestions to improve the manuscript:
1. The abstract does not clearly describe what exactly the study involves and its subsequent aims. Stating it is exploring an “early rehabilitation program” is ambiguous and fails to report what exactly is meant by “rehabilitation” and what physical areas it encompasses.
2. Ensure the tense is consistent throughout. Suggest not using the term “was” to report previous literature in paragraphs between lines 67 and 75.
3. “Advanced treatment technology” lines 49-50 is ambiguous. Does this mean treatment of advanced cancer?
4. Suggest delete word “cosmetics” line 51 and replace with “appearance”
5. Suggest exclude “..along with the usual sequelae of oral cancer treatment” Line 52– it is ambiguous and does not add to the sentence.
6. Line 54 “.. advanced stage..” please describe if it is meant advanced stage of cancer or treatment. Also provide an example of donor site-related impairments.
7. Remove “.. sublevel IIB neck dissection..” lines 61-62 for same reason given in point 5.
8. Suggest delete paragraph starting with “Additionally.. “ line 62 as the reason that spinal accessory nerve sparing procedures may experience ANSD is not related to innervation of trapezius from C2-4 cervical branches, but rather that varying degrees of nerve injury can occur with an intact nerve (i.e. neurapraxia, axonotmesis) Please refer to refs 4-6 in your manuscript.
9. Paragraph from line 67-75 lacks clarity. This paragraph lists an entire range of possible post-treatment sequelae which aren’t all relevant to the rehabilitation program of the study (for example “.. increased xerstomia..” line 73, “.. speech and social eating” lines 73-74). Instead this section needs to outline what specific physical issues relating to health-related quality of life the study is aiming to address. Suggest replace “.. residual..” in line 68 with “treatment-related morbidity”. Also in this line delete remainder of sentence from “.. ANSD..” and replace with “physical issues”.
10. Colloquial terms in line 76 “In light of..” and “.. we decide”.

Experimental design

I commend the authors for the effort associated with recruiting such a challenging group of research participants. There is a broad range and large number of outcome measures that were assessed during the study, which overall were well supported with references.
Unfortunately the study design has major design flaws, which mean that subsequently the validity of the findings is inaccurate. These substantial methodology issues are as follows:
1. The aim is not specific enough in lines 80-85. The term “.. early-intervention program..” needs further description. There is no outline of what therapy discipline the program is entailing and what specific issues. If the program is multi-faceted, then this needs stating.
2. Further description of participants needed such as location and setting, method of recruitment, area of tumour (please add this to Table 1).
3. Poorly defined as to what exactly were the physical limitations encountered by the study population in lines 108 “.. the problems suffered as a result of the surgery..” and again in line 129 “.. impairment suffered from the surgery or radiation therapy..” .
4. The interventions are poorly described in the Methods section, and there is no specific dosage as to the rehabilitation received which is essential when investigating any rehabilitation intervention. For example what does “.. pain management, edema control etc..” lines 109-110 entail? What is “.. electrical stimulation..”?
5. There are no supporting references to justify the huge array of therapy undertaken. Additionally, there is poor evidence and clinical reasoning for this program.
6. No mention of assessor blinding, means there is a risk of bias with this study if the assessor was aware of what follow up time point participants were at the time of assessment.

Validity of the findings

As outlined above, the methodological issues mean that the conclusion of the study cannot be support.
1. Lack of a control group means that the study lacks the internal validity to aim for evaluation of outcomes due to a rehabilitation program because of potential confounders such as natural recovery. This and methodological flaws mean that the study’s conclusion that rehabilitation is “necessary for oral cancer survivors” is inaccurate.
2. Recommend providing data as to the type of surgery and tumour site (eg. Tongue, Mandible) in the Results section and corresponding Table.
3. Please provide reason(s) why 13 participants decided not to enrol.
4. The drop-out rate of 34% reported is reasonably high. Data contained in Table 1 reports that 37 participants out of the 65 included participants received radiation therapy. With 30 out of that number dropping out due to the “discomfort caused by RT”, as the CONSORT diagram in Figure 3 indicates, this would mean that for 81% of participants receiving radiation therapy this rehabilitation was not feasible. This result would be an interesting outcome to explore and describe, particularly as there is limited evidence as to the effect of rehabilitation during radiation therapy.

There are some interesting findings from this study such as the feasibility of a generalised rehabilitation program during radiation therapy and changes on a large range of outcome measures (and physical issues) over a 6 month time period. Furthermore, the data collected on return-to-work could be analysed further to potentially provide any association between return-to-work status and vocation type (as per data collected in Table 1) which may be of predictive value and therefore clinically useful information. If any data was collected as to reasons for not being able to return-to-work, would again be of clinical benefit. I suggest the manuscript be re-written to change its focus towards this data, of which a single-arm pilot study can provide valid and clinically-useful data.

Reviewer 2 ·

Basic reporting

The outcome measures are well chosen and relevant. And I want to commend the authors for the long follow-up time in this study. If there is anything new in the intervention itself (which there doesn’t have to be!) compared to ‘normal practice’ that could be more clear.
Figures are relevant and of high quality – the photos are very informative.

The following sequences need linguistic editing:
Line 71 – ROM – please note, that ROM is not explained until line 110.
Language in line 76-77
Line 84 instead of ”the early and the advanced period” I suggest “the early and the advanced stage population eligible for surgery” otherwise it is a quite unclear what you mean.
Language in line 388-89.

Experimental design

Methods are very well described with sufficient detail & information to replicate. Well done!

Validity of the findings

As far as I can see the raw data from the questionnaires is not provided – or perhaps just not labelled in a comprehensive manner?

33 out of 98 discontinued and the intervention is quite demanding. I believe some thoughts on feasibility are required? If you have the possibility to present whether there were any differences in baseline-characteristics of the dropouts compared to the ones who completed the intervention that might help the reader in deciding who the intervention is most suited for.

---

## Round 0.2 · Major Revisions

· Academic Editor

Major Revisions

The revised manuscript adds substantial detail to the specification of the intervention.

It isn’t clear why the patients who did not complete follow-up have been excluded from this analysis. This will potentially bias the results. The paper should summarise all patients who received the intervention (if no outcome data was collected we should at least be provided the baseline characteristics of the patients who dropped out so we can ascertain the context / potential for bias).

There is still no recognition or discussion of multiple comparisons. This is a major limitation of the study analysis. There are 5 shoulder outcomes (plus the ratio endpoints and quality of life) and there have been tests made at multiple time points - many of the “statistically significant” results could be false positives. If all exact p-values are not reported the total number of tests undertaken must be clearly specified.

It still isn’t clear what the results in table 4 and 5 are based on – is this a summary of the raw data or does this come from the GEE. What is B? Reporting (p =0.000~0.033) is not informative and should be made inline if not reported in the tables.

There are a number of inconsistencies in the numbers reported e.g. dropout rate is not reported as both 34/35%. The manuscript requires a thorough review and edit for English language before it could be accepted for publication.

Note: there is still one “P= .000”.
Table 1 only lists 4 p-values but there are 13 variables.
The ‘number at risk’ provided do not make sense.

---

## Round 0.3 · Minor Revisions

· Academic Editor

Minor Revisions

Thank you for confirming that all outcomes detailed in the methods section were evaluated. Given the substantial number of outcomes (and small sample size) it is important that this is very clear to readers – the number of tests/models fitted should be reported. I count a total of 15 outcomes (4 ROM, 7 strength related, 2 oral health, 2 quality of life, 2 functional). Additionally, the 28 (15+13) quality of life sub-scales have been evaluated in three ways (overall and by subgroup). In the context of 65 patients, there are concerns about the validity of parameter estimates.

The authors confirm that only the “statically significant results” are stated in the Results section of this article or in the Tables. It is the number of non-statistically significant results which are critical to the interpretation of this many tests. I.e. are “significant” results seen more than 5% of the time?

There are three possible remedies: one is to ignore some analyses (this is not recommended now that they have been conducted), another is to report shrinkage estimators (however this would further complicate the analysis), the last is to clearly report all analyses and be upfront about the large scope for false positive results.

The authors already present the majority of the models undertaken. The interpretation of this study must be tempered within the context of so many outcomes evaluated on so few patients. An additional limitation paragraph discussing this issue is required.

Additionally:

It is still not clear what the p-values in tables 2 and 3 represent. E.g. are they tests of a time effect?

It is not reported what correlation structure was used for the GEE models.

Please explain why baseline was not used as the reference time point in the GEE models (it is usually clearer to report changes relative to baseline)?

ROM is not defined

---

## Round 0.4 · accepted · Accept

· Academic Editor

Accept

I am happy to Accept your article.

Reviewer 1 ·

Basic reporting

The points raised in the Reviewer comments have been sufficiently addressed and changed.

Experimental design

The points raised in the Reviewer comments have been sufficiently addressed and changed.

Validity of the findings

The points raised in the Reviewer comments have been sufficiently addressed and changed.

Additional comments

The points raised in the Reviewer comments have been sufficiently addressed and changed.